# Contextualizing Physical Data in Professional Handball: Using Local Positioning Systems to Automatically Define Defensive Organizations

**DOI:** 10.3390/s22155692

**Published:** 2022-07-29

**Authors:** Brice Guignard, Claude Karcher, Xavier Reche, Roger Font, John Komar

**Affiliations:** 1Centre d’Études des Transformations des Activités Physiques et Sportives UR 3832, UFR STAPS, University of Rouen Normandie, 76000 Rouen, France; brice.guignard@univ-rouen.fr; 2Mitochondria, Oxidative Stress and Muscular Protection Laboratory (EA 3072), Faculty of Medicine, University of Strasbourg, 67081 Strasbourg, France; claude.karcher@gmail.com; 3European Centre for Education, Research and Innovation in Exercise Physiology (CEERIPE), 67000 Strasbourg, France; 4Olympic Sports Performance Center (CREPS), 92291 Strasbourg, France; 5Sport Performance Area FC Barcelona, 08016 Barcelona, Spain; xavier.reche@fcbarcelona.cat (X.R.); rogerfontribas@gmail.com (R.F.); 6INEFC Barcelona Sport Sciences Research Group, National Institute of Physical Education of Catalonia (INEFC), University of Barcelona (UB), 08007 Barcelona, Spain; 7School of Health Sciences, Tecnocampus, Pompeu Fabra University (UPF), 08302 Barcelona, Spain; 8Physical Education and Sports Sciences, National Institute of Education, Nanyang Technological University, 1 Nanyang Walk, Singapore 637616, Singapore

**Keywords:** LPS, indoor GPS, sensors, defensive organization, external workload

## Abstract

In handball, the way the team organizes itself in defense can greatly impact the player’s activity and displacement during the play, therefore impacting the match demands. This paper aims (1) to develop an automatic tool to detect and classify the defensive organization of the team based on the local positioning system data and check its classification quality, and (2) to quantify the match demands per defensive organization, i.e., defining a somehow cost of specific defensive organizations. For this study, LPS positional data (X and Y location) of players from a team in the Spanish League were analyzed during 25 games. The algorithm quantified the physical demands of the game (distance stand, walk, jog, run and sprint) broken down by player role and by specific defensive organizations, which were automatically detected from the raw data. Results show that the different attacking and defending phases of a game can be automatically detected with high accuracy, the defensive organization can be classified between 1–5, 0–6, 2–4, and 3–3. Interestingly, due to the highly adaptive nature of handball, differences were found between what was the intended defensive organization at a start of a phase and the actual organization that can be observed during the full defensive phase, which consequently impacts the physical demands of the game. From there, quantifying for each player role the cost of each specific defensive organization is the first step into optimizing the use of the players in the team and their recovery time, but also at the team level, it allows to balance the cost (i.e., physical demand) and the benefit (i.e., the outcome of the defensive phase) of each type of defensive organization.

## 1. Introduction

Handball is an intermittent high-intensity team sport that involves short, high-intensity actions such as accelerations, decelerations, changes of direction, sprints, jumps and many collisions [1]. Handball is an invasion game opposing two teams of seven players (six field players and one goalkeeper) during 2 periods of 30 min. A previous review [1] have shown that elite handball player covers from 2757 to 4964 m during a match. This distance is composed of various forms of displacement such as side steps, backward running, high-speed running (HSR) and sprints [2,3]. 

Playing roles influence largely match demands [1,4] found that center backs and wings (4040 ± 1007 m and 3903 ± 1224 m respectively) cover largely more distance than the other players. Wings and center backs were more involved in HSR action both in distance (410 ± 193 m and 243 ± 130 respectively) and in number (39.6 ± 18.2 and 34.9 ± 18.1). When time motion was studied [5], it was shown that wing players spent significantly more time standing and walking than the other players during matches. Technical demands were also playing position-dependent; pivots received and gave up to largely more contacts (58.9 ± 20.3 light contacts and 11.6 ± 3.2 high-intensity contacts [6] while pivots perform significantly more duels (36.4 ± 16.2) and backs significantly more jumps (19.1 ± 5.2) [7].

In the defensive phase, six field players and an unviolated area defended by a goalkeeper are protecting the goal. These constraints induce that 73% of the ball possession is played in a stabilized attack [8], as a result, most of the playing time is spent in these phases. Some recent studies have analyzed the difference between the offensive and defensive phase in elite male international players [5,9], however inconsistent results were found for some of the variables (e.g., total distance, walking distance). In both studies, data indicate that players in defense covered a greater distance in high-intensity zones (23% in high-intensity running and 30% in sprinting) than in offense. To our knowledge, all those studies did not specifically consider the stabilized defensive phase. They merged the recovery phases (with a lot of high-speed running and sprints) with the stabilized attack. This choice may have biased the result.

It is worth noting that because players’ rotations are unlimited in handball some players are specialized in the defensive or in the offensive phase. To our knowledge, only one study in elite female handball players took this into account despite their importance in the result in terms of external workload [10]. Moreover, no previous study has investigated the effect of the defensive formation on the match demands in handball. In soccer, some studies have investigated the effect of different defensive organizations and found substantial differences between defense types [11,12]. In handball, the spatial organization is heterogeneous, the two most used formations are the 1–5 (a first line of five players the nearest from the goal and a player in front of this first line) and the 0–6 (all players in the same line from the goal). The necessity for a team to master different defensive organizations can bring a tactical advantage by using the weaknesses of the attacking team, for instance, if the back players are short, 0–6 defensive organization may be more effective in preventing them to shot. On the other side, if the back players of the attacking team lack of mobility, a 3–3 defensive organization (a first line of three players the nearest from the goal and three players in front of this first line) may lead to a lot of interceptions. However, as those different defensive organizations seek to exploit different defensive areas (i.e., more or less depth), it is very likely that in handball defensive organizations also affect many sides of the match demands (e.g., external and internal load, technical demands), as it was observed in soccer.

A growing number of elite teams are equipped with a local positioning system (*Tracking*, s. d.) and many studies used this technology to study match demands [4,5,9]. Automatic detection of the defensive formation could be very useful both to help coaches to design adapted training content and in a research context. The aims of this paper were (1) to develop an automatic tool to classify the defensive organization of the team and check its classification quality, and (2) to quantify the match demands per defensive organization, i.e., defining a somehow cost of specific defensive organizations.

## 2. Materials and Methods

### 2.1. Data Sample

The data were provided by a Spanish first division handball club and cover 25 matches of several competitions over the 2019–2020 and 2020–2021 championships. This club finished in the top 5 teams over these two years. From those two seasons, all the games that were played at home and when all the players were equipped with the LPS system are included in the data sample. Over those 25 games, 30 different male handball players appeared in the team (average ± SD for height: 1.90 ± 0.06 cm; weight: 91.9 ± 12.5 kg and age: 26.3 ± 3.9 years old). For each single match, the group was composed of 12 to 16 players from the pool of 30. The data correspond to X;Y positions of the players on the handball court as well as the corresponding television video broadcasts of the matches.

### 2.2. Data Collection

Positional data of the players: The position data were collected by the professional team staff members from the WIMU PRO inertial measurement system (RealTrack Systems S.L., Almería, Spain). The sensors are placed on the back of the athletes between the two scapulas at the level of the C7 vertebra. They are positioned in a special neoprene waistcoat in order to keep the sensor as fixed as possible.

The WIMU PRO inertial measurement system has been the subject of several studies to ensure its validity and reliability. Three of them have been published very recently, from 2019 to 2020: [13,14,15]. The various studies consisted of the performance of different linear and non-linear exercises at different intensities. The reference used in these studies is the ground marking of the pitch and a geographical mapping system via video recordings of these sessions. The studies analyzed the system with a 6-antenna configuration [13] and then with an 8-antenna configuration [15] in order to see if there were differences between the configurations. The observed variables are the X and Y positions of the inertial sensors. Both studies validate the use of the LPS and the estimation of positions is considered accurate. The use of 8 antennas has better spatial accuracy than the other configuration (*n* = 6 antennas) with a 3 cm error instead of 5.5 cm, but both configurations are usable in the positional data estimation. Furthermore, the comparison between the positional data estimation from two different units is good (CV < 1%) so this system has good inter-unit reliability. Each sensor weighs 70 g and its dimensions are 81 × 45 × 16 mm (height/width/depth).

Each sensor has a number and is affiliated with the same player for the entire season to limit problems that may exist due to inter-unit reliability of the sensors. The configuration used in this data collection is six antennas positioned around the field with the (0;0) coordinates corresponding to a corner outside the field (Figure 1, left panel). As mentioned in the study by Bastida-Castillo et al. [13], the way this LPS UWB system works is that it is composed of two subsystems: the reference system which is composed of the antennas that act as radio transmitters and receivers and the sensors carried by the participants. The system obtains the position of the sensor via triangulation of its position between the antennas and thus obtains its position in X and Y.

### 2.3. Data Validation Set

Two handball matches were randomly extracted from the sample and sequenced with Dartfish (Fribourg, Switzerland) using a specially designed sequencing panel. The panel was used to record the actions carried out by the professional handball team, which wears the WIMU inertial units: the start of the match, the number of attack and defensive phases, the number of 2-min subdued, the number of 7 m shots (penalty throws), the number of sequences played without a goalkeeper, the presence of time-outs (requested by the coaches or imposed by the presence of half-time), as well as the transitions such as the engagements, were identified. If an event such as a penalty throw occurs during an attack, the attacking phase is not split into two distinct phases: it remains as only one attacking phase. Similarly, if an event such as a yellow card occurs during a defense, only one defending phase is considered. Within these sequences, qualifiers could be added such as the type of defense (i.e., 3–3, 2–4, 1–5, 0–6). Duration of attacking and defending phases were computed and will be compared to those obtained from the automatic data treatment.

### 2.4. Automatic Data Treatment

The positional data were transformed from the WIMU PRO software (SPRO, Real- Track Systems, 2018, version 964 [16]) format to a comma-separated value (.csv) format. The version of this software corresponds to the number 964. The exports are composed of X and Y data obtained by the LPS system but also data of a 3-axis accelerometer, a 3-axis gyroscope and a 3-axis magnetometer (data from the WIMU inertial sensors). These later data were not considered in the present study, which is based exclusively on the X and Y positional data. The data treatment was performed on a custom-made Matlab algorithm (The MathWorks^®^, Inc., Natick, MA, USA, r2016b).

#### 2.4.1. Step 1: Data Import

Importation of the WIMU data from the csv. files player by player. Concatenate the data saving the X and Y positions for all players and the time of each registered data. The sampling frequency of the positional data ranged from 17 to 18 Hz (see step 3).

#### 2.4.2. Step 2: Goalkeepers Identification

The data corresponding to the performance of the goalkeepers (GK) were differentiated from the other on-field players. This was performed from a heatmap of the positions of all the players. The players with the highest density in the 6 m zone were retained as a goalkeeper. Generally, two players are identified as GK during one handball match.

#### 2.4.3. Step 3: Data Resampling and Clock Format

The frequencies of each WIMU sensor in the LPS system were not constant and ranged from 17 to 18 Hz. Plus, they were not turned on at the exact same clock before the match start (we transform the initial format from hours, minutes, seconds and milliseconds –hh:mm:ss.ms– to data expressed in seconds only). Therefore, a resampling was necessary to adopt a common frequency for all WIMU. This frequency was equal to 16.67 Hz, which corresponds to a time increment of 0.06 s between each registered positional data).

#### 2.4.4. Step 4: Translation of Coordinate System

For greater clarity in the presentation of the data, the positioning reference, initially defined outside the boundaries of the field, has been repatriated to the centre of the field (on the median line, perfectly in the centre of the field on X and Y axes). To do this, we used the barycentre of the positions of the two GKs when they were in front of their goal (Figure 1, right panel). When the barycentre of the data was obtained, the set of positions of all the players was translated from their initial positions to the actual position they should occupy on the field. Thus, the extreme values of the field corresponded to −10 and +10 m on the *X*-axis (field width) and −20 and +20 m on the *Y*-axis (field length). In order not to lose data that could be due to reconstruction errors of the LPS system, we applied some tolerances: when the data on the *Y*-axis are between −15 and +15 m, only the data on X between −9 and +9 m are kept (the others are deleted). When the Y data is beyond the previous limits (i.e., the wings), then the X data from −11 to −10 m and from +10 to +11 m are taken into account and automatically reported to −10 and +10 m respectively. This would prevent wing players from being considered outside the field when they are near the field boundaries. From this cut, all points outside the field boundaries were no longer considered for the analysis.

#### 2.4.5. Step 5: Match Data Cleaning

To automatically detect the official ceremony prior to the match, we look for the moment when more than 10 players were present in a 2 m band near the center of the field on the *X* axis (from −2 to 0 m). All data registered before this moment were removed from the analysis. To automatically detect the presence of a halftime break, we created a binary algorithm that is based on the positions of all GK relative to the center line of the field on the *Y* axis. So, when the algorithm detects an abrupt change in the data (e.g., from one field side to the other on the *Y* axis), we consider it to be data from the halftime break. From that detection, a manual selection of the precise starts and ends of both halves of the match was performed, since only one GK should be on the field. For the start of the match, this rule was added to the fact that at least seven players should be in the field limits described in Step 4.

To facilitate the next steps of the automatic data treatment, positional data of one half were transformed to be equivalent to the other. Thus, on the *Y* axis, negative values of position data will be associated with defensive phases, whereas positive values will be associated with attacking phases.

#### 2.4.6. Step 6: General Match Analysis

Attacking and defensive phases

When at least one player is between 12.5 to 0 m of the opposing team’s goal line (*Y* axis > 7.5 m), the team is playing an Attack (ATT). For the defense detection, we based our computations on the positions of the on-field players. If at least 5 of those players are positioned from 0 to 12.5 m of their own goal line on the length of the field (i.e., less than −7.5 m on the *Y* axis), the team is playing defense (DEF). To avoid errors based on too many detections, it is not possible to observe a phase occurring less than two (2) seconds after the end of the previous phase. Furthermore, ATT or DEF cannot last less than two (2) seconds.

Defensive organization

The team is considered to play a defense system if 1. the duration of the DEF phase and 2. the time gap between two consecutive DEF are higher than seven (7) seconds. This duration is the minimum duration for a potential fastbreak performed by the team.

From that definition, several defense organizations are categorized from the Euclidean distance (ED) computed between each on-field player position to the center of their goal line (coordinates (X;Y) equals (0;−20)). If the team is in a DEF phase, and all players present an ED from the player position to the center of the goal line less than 9 m, the defense is qualified as ‘0–6′. This denotes six players near the goal area line. If the team is in a DEF phase, and 1 player has an ED superior to 9 m, the defense is qualified as ‘1–5′. This denotes 5 players near the goal area line and one player outside the 9 m zone. If the team is in a DEF phase, and 2 players have an ED superior to 9 m, the defense is qualified as ‘2–4’. This denotes the first form of open defense. If the team is in a DEF phase, and 3 players have an ED superior to 9 m, the defense is qualified as ‘3–3’. This denotes a second form of open defense.

Team timeout detection

If more than nine (9) players are detected on the defending side of the field, with a position close to the substitutes’ bench side (from the side of the field to a line 3 m away from the center of the field on the *X* axis), with a duration of around one (1) minute (exact range is comprised from 50 to 90 s). If that detection occurs, the attacking or defending phase that surrounds the team timeout was split into two distinct phases. This methodological point offers the possibility to detect a change in the way the team attacks or defends after a team timeout.

Two minutes exclusion detection

If less than 7 players are on the field, and in a time comprised from 120 s to 120 s + 20%. If the detection is longer than the latter limit, an additional 2 min embedded in the first is/are detected. If less than one (1) second is observed between two players’ inferiorities, the inferiority qualification is maintained.

#### 2.4.7. Step 7: Automatic On-Field Player Roles Detection

Player roles were determined from their behaviors during the ATT phases only. Players involved in less than 10% of the total attacking time of the team were not considered. Detection rules are listed below, starting with the wing players then the line players, and finally back players.

Wing players detections

During an ATT, wing players are detected as the players with the highest density of positions in an area from each sideline to 2 m inside the field, when they have a position beyond 15 m on the attacking side (from the goal line of the opposing team to 5 m inside the field on the *Y* axis) (Figure 2 left panel).

Line players detections

During an ATT, line players are detected as the players with the highest density of positions in an area from −5 to + 5 m to the center line on the length of the field (i.e., *X* axis), when they have a position beyond 7.5 m on the field of the opposing team (from 7.5 to the goal line of the opposing team on the *Y* axis) (Figure 2 middle panel).

Back players detections

During an ATT, back players are detected as the players with the highest density of positions beyond 2.5 m to the midline of their own field (i.e., −2.5 m on the *Y* axis) and less than 12.5 m from the opposing team’s goal line (i.e., +7.5 m on the *Y* axis). Additionally, if the highest density of positions is detected either on the right or left side of the field (a band from the sidelines to 5 m inside the field, from 5 m from the opposing team goal line to 7.5 m to the mid-line of the field), the player was also classified as a back player. These zones are represented in Figure 2 right panel.

If the players do not satisfy any of these rules, they are no more considered for the rest of the analysis.

#### 2.4.8. Step 8: Physical Demands of Professional Handball

From each player’s positional data, the distances covered over time were calculated as the Euclidean distance from one coordinate (X;Y at n) to the subsequent one (X;Y at n + 1). Those distances were computed for the whole team at the scale of the whole match, and for each half during the four defensive organizations. Additionally, distances covered by wing players, back players and line players during the zone defenses were also computed. A similar methodology to the one of Alt et al. [17] was used to process the data. First, a numerical differentiation over time was performed from the coordinates of each player on the field to obtain their instantaneous speed data. Then, to minimize noise, the speed data were passed through a second-order Butterworth low-pass filter with a 1-Hz cut-off frequency [17]. A second numerical differentiation was performed from those data to estimate the player’s deceleration over time. To detect the start and end of the stand, walk, jog, and sprint sections, the thresholds of Büchel et al. [18] were applied. Therefore, we considered the following zones as standing from 0 to 0.2 m.s^−1^, walking from 0.2 to 2.0 m.s^−1^, jogging from 2.0 to 4.0 m.s^−1^, running from 4.0 to 5.5 m.s^−1^ and finally sprinting when the instantaneous speed was over 5.5 m.s^−1^. The number of sprints corresponds to the number of times a player crosses this sprint velocity threshold. According to Stevens et al. [19], a deceleration from running to a standstill started when acceleration fell below −0.2 m.s^−2^ in soccer, so we replicated the same rules for the present study in handball to compute the number of players’ decelerations. The distance covered, the number of sprints, accelerations and decelerations performed by the whole team, and by each group of players (i.e., wing, back and line players) for each defense system were computed.

We then analyzed the defensive intention as the defensive organization that was the most observed during the first seven (7) seconds (i.e., during the start of the stabilized defensive phase) of the defensive phase. Prior to this analysis, we automatically removed two (2) seconds at each beginning and at each end of the DEF phase to remove transitions between ATT and DEF phases and the DEF and ATT phases.

Each deviation from the intention and the performed defensive phase would denote an adaptation relative to the attacking team.

## 3. Results

A total of 25 handball matches were included in the present study. It corresponds approximately to the analysis of 3000 ATT and DEF combined.

### 3.1. Data Validation Set

Duration of the ATT and DEF phases were computed and compared from the Dartfish sequencing and the automatic Matlab algorithm. Details of the comparison between the two methodologies and potential explanations for non-detection are presented in Appendix A (Table A1, Table A2, Table A3 and Table A4). On average, during the first match, absolute differences of 4.7 ± 11.9 s (*n* = 30 and 28 for Dartfish and Matlab in the first half) and 9.4 ± 20.5 s (*n* = 29 and 32 for Dartfish and Matlab in the second half) were observed for attack durations. These values were very similar to those observed for the second match (4.1 ± 10.5 s and 9.1 ± 20.4 s for each half). The quantity of ATT was *n* = 30 and *n* = 28 during the first half, and *n* = 29 and *n* = 31, for Dartfish and Matlab programs, respectively. The duration of absolute differences for defensive phases was 11.8 ± 14.9 s (*n* = 30 and 35 for Dartfish and Matlab; first half) and 15.7 ± 22.3 s (*n* = 30 and 33 for Dartfish and Matlab; second half). During the second match and its first half, the number of DEF phases was *n* = 29 for both detection techniques (absolute difference of only 5.6 ± 5.2 s), whereas it raised to 11.8 ± 17.5 s during the second half (*n* = 30 and 35 for Dartfish and Matlab).

### 3.2. Defensive Intention vs. Actually Performed Defense

On average for the 25 matches, the 1–5 DEF type occurred more than the others, both for the defensive intentions and the actually performed defenses (Figure 3). 1–5 DEF type was also the only defense that was performed more frequently than it was intended to by the team at the beginning of the DEF sequence. All other DEF types (0–6, 2–4, and 3–3) were on average intended to be performed more frequently than it was actually the case. Less than 5% of defense systems were not classified as one of the four defensive organization types defined in the present study.

### 3.3. Physical Demands as a Function of Defensive Organization and Players’ Role

The physical demands associated with the defensive organization are presented in Table 1 and Table 2. Based on the intended and actually performed defensive classification, the physical demand of each type of defense are presented per 10-min of play in each specific type of defense for easier comparison but in two different ways, based on the intended defense (Table 1) and based on the actually performed defensive organization (Table 2).

## 4. Discussion

The aim of this paper was to go further than a breakdown of match demands by players’ roles by looking at the cost of specific defensive organizations implemented during the game on the match demands. Using the raw data from a local positioning system, we aimed at developing an automatic tool to classify the defensive organization of the team and check its classification quality, and quantify the match demands per defensive organization, i.e., defining a somehow cost of a specific defensive organization.

Starting from the raw data from the local positioning system, multiple steps were necessary to get a clear view of the match context, for instance removing the players who stand on the bench is a key step necessary prior to defining any match information (e.g., handball defensive or offensive activity), whereas this step is usually not considered when looking at the global match demands of the players [4]. Indeed, handball is a sport with unlimited, and high rotations between players (i.e., players come in and out of the game very easily and frequently), similar to ice hockey, and while they usually sit on the bench when they are out therefore not drastically impacting the match demands numbers (i.e., no change in sprinting distance covered, etc…), those rotations do impact the detection of the defensive/offensive organization [4]. Defining the context of the player’s activity, specifically when using sensors (i.e., without any information on the players from the opposition team nor on the ball location) necessitates a drastically deeper processing of the data than quantifying the global match demands. After this cleaning, the rule based algorithm tested in the present paper provided decent automatic detection of the context of the players’ activity, starting from the detection and duration of the defensive and attacking phases. Although it does not differentiate a stop in the game due to a 2-min suspension compared to a stop due to a player injured (see Annex A), it is still possible to detect those situations as *anomalies*, i.e., no stabilized attacking or defensive phases and therefore exclude them when computing the match demands of specific context of play (e.g., the physical demand of a 1–6 defensive organization). Eventually, compared with a regular coding using video analysis software, the algorithm can detect the stabilized attacking and defensive phases with high accuracy including the duration and exclude what is not considered such a phase in the game without detecting precisely what kind of situation is excluded though (e.g., 2-min exclusion, player injured, time out), with a total of 6 errors of detection for the two games we analyzed (i.e., a total of 266 attacking and defensive phases).

From there, the current analysis focused on the stabilized defensive phases with the detection of the defensive organization (i.e., the geographical organization of the players on the court), based on the definition of those different organizations (1–5, 0–6, 2–4 or 3–3) without merging it with the transition phases (e.g., [5,9]). Interestingly when dealing with the defensive organization, the highly dynamic and interactive nature of handball (and generally team games) makes the characterization of the organization defensive sensitive to the opposition. In other words, the intention of a team to defend with a specific organization can easily evolve based on the moves and activity of the attacking team (i.e., for which there is no data as the sensor are worn only by our own team). In that sense, what the team initially intends to do at the start of the defensive phase can be seen potentially only at the start of the phase as it will be disrupted by the attacking team (e.g., an attacker entering the defense as a second pivot, or large moves or attacking players without the ball). Comparing the actual defensive organization in its full dynamics to the original intention of the team at the start of the defensive phase shows that indeed the intended organization is not similar to what actually happens within the defensive phase, but mainly for the 1–5 and 3–3 intended defense organizations. On the other side, 0–6 and 2–4 defensive organizations appeared to be very stable and independent from the opposition activity. Potentially, 1–5 and 3–3 defensive organizations could be seen as being easier to disrupt when you attack against them, but they could also simply be more adaptative in the way they are implemented. For instance, a 0–6 defensive organization is usually implemented as it is very stable and robust against offensive perturbation. By extension while looking at the match demands in regard of the defensive organization, the players’ physical activity in terms of distance covered at different speed intensity does differ between players’ roles as expected [1,4], but also can differ between the consideration of the defensive organization (i.e., between the original intention at the start of the defensive phase or the actual geographical organization of the players), specifically when using a 1–5 or 3–3 defensive organization. In this case, adding the defensive context to filter the global match demands allows to quantify the cost of the defense, for instance, the wingers will show higher demands (i.e., walk, jog and run) when the team is defending in 1–5 compared to a 0–6. However, the distinction between the intention and the actually implemented defensive organization has to be considered also, taking into account the dynamic nature of a defense phase. In the present case, both versions were computed and probably not one unique solution would be ideal. Eventually, to quantify the cost of a specific defensive organization, considering that the moves and activity of the attacking team players depend on how the defensive team is organized at the start of the defensive phase (i.e., so depend on the defensive intention), which in return will impact how the defensive team will have to adapt its own defensive organization, it seems relevant to use the defensive intention to label the organization of the entire defensive phase (i.e., regardless on how the players move and adapt this initial organization in the course of the defensive phase, the following match demands is assigned to the original defensive organization cost). On the other side when quantifying the cost of a specific defensive organization, coaches and analysts could prefer to qualify the defensive organization strictly as it is at each instant of the defensive phase (i.e., if one player has to move up from a 0–6 defensive original set-up, it becomes a 1–5 and the following match demands is assigned to the 1–5 defensive organization cost).

## 5. Conclusion and Limitations

The objective of this analysis was to develop an algorithm that can bring context into the quantification of the physical demand of a handball match. Using those sensor data seems a promising way in bringing some contextual handball knowledge into the match demands analysis. We showed that it is possible to effectively derive some qualitative information about the game like identifying defensive and attacking phases and even identifying the defensive organization. From there, quantifying for each player role the cost of each specific defensive organization is the first step into optimizing the use of the players in the team, but also which type of defensive organization the team should use.

Unfortunately, LPS systems provide only the position of the players wearing them, and the team analyst does not access the data of the opponents. In the same idea, no information is given regarding the position of the ball from the LPS (although optical systems do provide that information, they require intensive and time consuming post-processing). In that sense, investigating the defensive organizations using the LPS data was indeed possible, however a limit of this approach is that no information about the interactions is provided (i.e., how the defense adapts to the attacking team moves, for instance, if a second attacker enters the defense to join the 6 m line). So if it is possible to observe a modification of the team in its defensive organization, it is not possible with this tool to assess the relevance of this as there is no information on the opponents. In other words, it is possible to detect what the team is doing in defense, but not why (i.e., it is not possible to judge the relevance of the defensive activity, whether it is “good” or not).

In addition, investigating the attacking phases without having any information on the ball position and defensive organization the attacking team is facing seems highly limited if not useless. Indeed, to judge the relevance of the way a team is attacking (e.g., specific players’ moves) requires knowing how the defense is organized.

To somehow approach such a level of analysis (i.e., how both teams interact), in the future the outcome or benefit of the defensive phase could also be identified in order to find “better” outcomes, for instance looking at the running speed of the defenders right after they move from defense to attack could help to identify fast breaks that are known to have a higher chance of scoring. In that sense, using a 3–3 defensive organization could lead to a higher chance of fast break following the defense which leads to a higher chance of scoring compared to a 0–6. However, a 3–3 defensive organization has a higher cost in terms of physical demand. Reaching this step could allow us to really balance the cost and the benefits of the different defensive organizations. Another future development would be to get the information from the algorithm live during the game (instead of post-processing), in order to really assist the decision-making of the head coach and strength & condition coach.

## Figures and Tables

**Figure 1 sensors-22-05692-f001:**
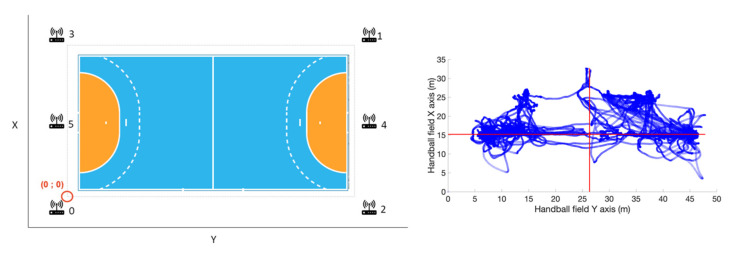
The six antennas configuration, positioned around the handball field with the (0;0) coordinates on the lower left corner (**left** panel). Corresponding positions reported on the field for the two GK (in blue, **right** panel), and the computed center of the field, that becomes the (0;0) coordinates system (crossing red lines).

**Figure 2 sensors-22-05692-f002:**
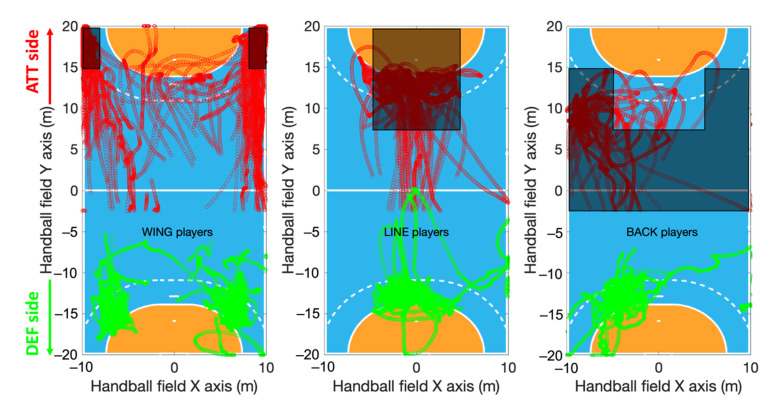
Detection of the players main roles based on their highest density of positions during ATT phases (in red). Classification of wing players occurs if the highest density of position corresponds to a black rectangle of 5 m long and 2 m wide on the wings (**left** panel). Classification of line players occurs in a zone of 10 m wide for 12.5 m long (black rectangle in the **middle** panel). Classification of back players occurs if the highest density is reported in the black zone of the field (**right** panel). As a supplementary visual indication, positions of each player during DEF phases were also depicted (in green).

**Figure 3 sensors-22-05692-f003:**
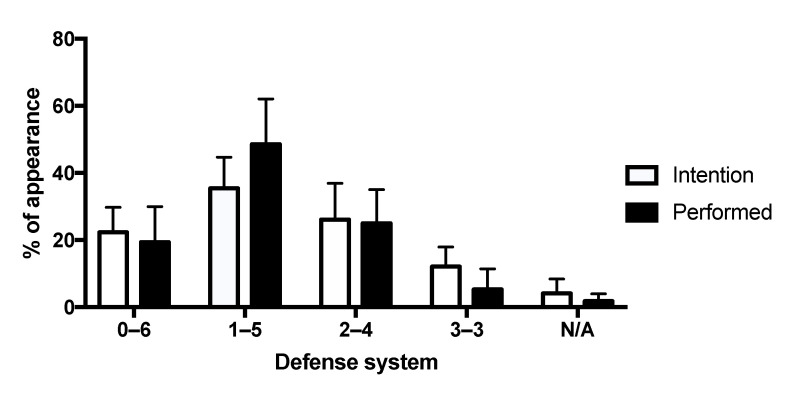
Mean ± SD percentage of appearance of each defensive organization (0–6, 1–5, 2–4, 3–3) for the sample of 25 matches included in the study. N/A corresponds to DEF that could not be classified into one of the four defensive organization according to our detection rules.

**Table 1 sensors-22-05692-t001:** Mean ± SD physical demands intention as a function of defensive organization and players’ role, normalized per 10-min of play.

Defense	Role		Stand (m)	Walk (m)	Jog (m)	Run (m)	Sprint (m)	Sprints (n)	Deceleration (n)
0–6	wing	M	0.00	722.42	65.75	9.88	10.83	7.28	360.00
		SD	0.01	100.43	30.37	15.56	29.43	25.85	45.00
	back	M	0.00	904.50	88.72	11.53	2.02	0.16	413.68
		SD	0.00	166.80	47.90	42.71	10.12	0.80	70.30
	line	M	0.00	277.30	32.37	0.65	0.35	0.04	127.08
		SD	0.00	116.13	18.92	1.39	1.73	0.20	56.29
	Total	M	0.00	1904.22	186.84	22.05	13.20	7.48	900.76
		SD	0.01	159.18	60.25	43.48	38.50	26.58	73.05
1–5	wing	M	0.00	724.46	75.84	9.39	3.86	0.36	353.20
		SD	0.00	90.60	29.30	11.30	7.72	0.70	40.75
	back	M	0.00	870.19	102.24	8.72	0.31	0.04	402.60
		SD	0.00	168.91	64.10	20.02	1.54	0.20	75.40
	line	M	0.00	262.25	27.63	0.46	0.00	0.00	122.12
		SD	0.00	126.54	19.79	1.24	0.00	0.00	55.68
	Total	M	0.00	1856.90	205.71	18.57	4.17	0.40	877.92
		SD	0.00	161.12	78.35	23.86	7.72	0.71	65.25
2–4	wing	M	0.01	741.16	83.77	9.26	2.32	0.12	359.76
		SD	0.02	89.67	39.64	9.38	8.30	0.44	45.61
	back	M	0.00	881.16	93.30	6.09	0.76	0.08	401.52
		SD	0.00	183.12	44.66	10.04	3.81	0.40	86.07
	line	M	0.00	252.03	26.32	2.07	0.00	0.00	116.44
		SD	0.00	157.76	21.40	6.51	0.00	0.00	72.47
	Total	M	0.01	1874.35	203.38	17.41	3.09	0.20	877.72
		SD	0.02	200.33	68.58	18.75	8.98	0.82	99.41
3–3	wing	M	0.00	614.55	69.59	9.33	1.25	0.24	292.76
		SD	0.00	174.25	42.25	13.45	2.99	0.66	83.25
	back	M	0.00	718.22	83.35	7.14	0.00	0.00	310.80
		SD	0.01	250.42	69.70	23.40	0.00	0.00	107.14
	line	M	0.00	220.01	19.85	0.00	0.00	0.00	103.36
		SD	0.00	139.48	17.89	0.00	0.00	0.00	65.39
	Total	M	0.00	1552.77	172.79	16.46	1.25	0.24	706.92
		SD	0.01	461.35	105.31	33.09	2.99	0.66	212.59

Note: M = Mean value; SD = Standard Deviation; 0–6, 1–5, 2–4, 3–3 are defensive organizations; wings, back, line are players’ roles.

**Table 2 sensors-22-05692-t002:** Mean ± SD physical demands performed as a function of defensive organization and players’ role, normalized per 10-min of play.

Defense	Role		Stand (m)	Walk (m)	Jog (m)	Run (m)	Sprint (m)	Sprints (n)	Decelerations (n)
0–6	wing	M	0.01	715.02	38.39	0.72	0.29	0.12	484.16
		SD	0.01	100.18	17.40	1.34	1.19	0.44	72.94
	back	M	0.00	907.47	72.45	0.38	0.00	0.00	563.44
		SD	0.00	164.51	39.91	1.38	0.00	0.00	101.16
	line	M	0.00	278.90	29.03	0.00	0.00	0.00	177.84
		SD	0.00	134.82	19.49	0.00	0.00	0.00	82.90
	Total	M	0.01	1901.38	139.88	1.10	0.29	0.12	1225.44
		SD	0.01	166.32	50.16	1.99	1.19	0.44	127.40
1–5	wing	M	0.00	754.80	70.04	6.12	1.04	0.36	559.20
		SD	0.01	82.91	22.97	5.90	2.12	0.70	63.68
	back	M	0.00	904.41	102.52	10.24	0.00	0.00	640.00
		SD	0.01	168.56	59.12	26.20	0.00	0.00	125.69
	line	M	0.00	271.09	28.08	0.49	0.00	0.00	196.24
		SD	0.00	126.12	12.47	1.21	0.00	0.00	92.32
	Total	M	0.00	1930.30	200.64	16.85	1.04	0.36	1395.44
		SD	0.01	147.68	70.19	26.15	2.12	0.70	136.43
2–4	wing	M	0.00	786.83	101.38	16.13	3.99	1.00	608.60
		SD	0.01	77.50	36.57	16.21	7.63	1.58	55.77
	back	M	0.00	931.81	112.64	12.94	0.10	0.08	700.04
		SD	0.01	165.93	62.18	35.32	0.50	0.40	142.33
	line	M	0.00	284.56	27.84	1.12	0.00	0.00	220.08
		SD	0.00	131.45	19.68	2.96	0.00	0.00	105.56
	Total	M	0.00	2003.19	241.86	30.19	4.09	1.08	1528.72
		SD	0.01	158.94	81.35	42.09	7.78	1.68	163.94
3–3	wing	M	0.00	819.82	136.98	21.64	12.00	3.84	647.40
		SD	0.00	87.57	54.56	33.49	15.91	4.55	72.65
	back	M	0.00	972.56	125.46	7.79	0.83	0.32	743.32
		SD	0.01	157.90	70.50	18.76	4.17	1.60	139.36
	line	M	0.00	300.26	37.49	2.69	0.72	0.24	243.16
		SD	0.00	132.90	38.54	5.66	3.58	1.20	128.48
	Total	M	0.00	2092.63	299.93	32.12	13.55	4.40	1633.88
		SD	0.01	190.28	107.69	39.92	15.89	4.80	211.59

Note: M = Mean value; SD = Standard Deviation; 0–6, 1–5, 2–4, 3–3 are defensive organizations; wings, back, line are players’ roles.

## Data Availability

Raw data from this study can be made available on demand by contacting the corresponding author.

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
