# Peer review of "Contextualizing Physical Data in Professional Handball: Using Local Positioning Systems to Automatically Define Defensive Organizations"

_sensors, 2022, doi:10.3390/s22155692_

Round 1

Reviewer 1 Report

Review a manuscript: Contextualizing Physical Data in Professional Handball: Using Local Positioning Systems to Automatically Define Defensive Organisations

Thanks for the opportunity to review this paper aiming to examine the to develop an automatic tool to detect and classify the defensive organization of the team based on the local positioning system data and check its classification quality, and to quantify the match demands per defensive organization, i.e., defining a somehow cost of specific defensive organizations. This is an interesting topic, however, some concerns exist regarding the rationale for this work and the context in which the data is discussed.

Specific Comments

Title:

Change the word “Organisations” to “Organizations”.

Introduction

Page 1, line 41: Change the word “travelled” to “covers”.

Page 2, line 71-74: In this sentence: “In handball, the spatial organization is heterogeneous, the two most used formation are the 1/5 (a first line of five players the nearest form the goal and a player in front of this first line) and the 0/6 (all players in the same line).” Suggestion to describe this sentence:  In handball, the spatial organization is heterogeneous, the two most used formation are the 1-5 (a first line of five players the nearest from the goal and a player in front of this first line) and the 0-6 (all players in the same line nearest from the goal).

Materials and methods

Page 4, line 154:  Check if it is 17hz or 18hz

Page 4, line 156:  Which is 000 after hh:mm:ss?

Page 7, line 274: In “Alt et al. 2020” insert the comma Alt et al., 2020

Page 7, line 278-280: Substitute “walking, jogging, running and sprinting” to walk, jog, run, and sprint. To be equal to table 1.

Page 7, line 280: Insert the values that are considered the accelerations.

Results

Page 8, line 329: Table 1. Change “Sprints (n)” to “Accelerations (n)”. And review the sprint numbers as they do not agree with the literature numbers.

Page 9, line 332: Table 2. Change “Sprints (n)” to “Accelerations (n)”. And review the sprint numbers as they do not agree with the literature numbers.

Reviewer 2 Report

I have carefully read the manuscript and my opinion is that the manuscript has a merit to be published in your reputable journal with some minor corrections. The manuscript is original, informative and readable. The authors tried to develop an automatic tool to classify the defensive organization of the team and check its classification quality, as well as to quantify the match demands per defensive organization, i.e., defining a somehow cost of specific defensive organizations. The structure of the abstract should be in the following order: the purpose of the study, method, results and short discussion with conclusions, so I would appreciate if the authors apply it as well as revise the abstract conclusion and offer potential readers not just what authors found but also to explained them what does it mean in short. On the other hand, the introduction is well written but I would appreciate if the authors more carefully describe the study sample in the method section as I prefer to see the handball players where they are coming from and their basic socio-demographic characteristics – this offers to further researcher to be able to compare the samples and consider some possible differences as a cause based on some socio-demographic differences. I would also prefer to see explanation why the authors selected this study sample, did they selected handball clubs and matches randomly (if not, need to be explained as it is possible that there is a serious methodological flaw if they favored some clubs). At the end, I have no amendments on results and discussion part but I would recommend to the authors to prepare the conclusion part in the following order: the main conclusions, the limitations of the study (more precisely) as well as recommendations for the further studies (it is very important to briefly elaborate it and highlight the most important notes, mostly due to the fact that this manuscript has an excellent practical application). Lastly, I would recommend you to accept this manuscript as soon as the authors revise it (I can review it one more time if necessary).

Reviewer 3 Report

Dear authors, first of all thank you for your submission to Sensors.

The article presents focuses on an interesting field of study within the field of handball and may be of special interest to researchers and coaches specializing in the field of handball. Even so, I believe that some points can be improved in order to enhance the quality of the manuscript.

Below you can find the specific comments.

Introduction

Line 37 - remove "team" at the beginning of the sentence

Line 74 - The idea exposed in this sentence needs to be further developed, because it is the fundamental basis of the study problem. Add support references., 

In addition, it is important to understand more accurately why the defensive phase was chosen to the detriment of the offensive phase

Methods and Results

This points are properly described good job.

Discussion:

Line 346 - Be consistent with the bibliographic reference format used. This occurs at other points as well, please check all references

Provide limitations for your study

Add a specific point with conclusions for future lines of investigation

Round 2

Reviewer 3 Report

Dear authors,

thank you for the answer.

The authors seek to comply with all the recommendations presented, and in this sense I have no further reservations regarding the study.